# Effects of Functional Depletion of *Doublesex* on Male Development in the Sawfly, *Athalia rosae*

**DOI:** 10.3390/insects12100849

**Published:** 2021-09-22

**Authors:** Shotaro Mine, Megumi Sumitani, Fugaku Aoki, Masatsugu Hatakeyama, Masataka G. Suzuki

**Affiliations:** 1Department of Biosciences, Nihon University, 3-25-40 Sakurajosui, Setagaya-ku, Tokyo 156-8550, Japan; 0848223025@edu.k.u-tokyo.ac.jp; 2Division of Biotechnology, Institute of Agrobiological Sciences, NARO, Owashi, Tsukuba 305-8634, Japan; sumikasashima@affrc.go.jp; 3Department of Integrated Biosciences, Graduate School of Frontier Sciences, The University of Tokyo, 5-1-5 Kashiwanoha, Kashiwa 277-8562, Japan; aokif@edu.k.u-tokyo.ac.jp; 4Division of Applied Genetics, Institute of Agrobiological Sciences, NARO, Owashi, Tsukuba 305-8634, Japan; sawfly@affrc.go.jp

**Keywords:** *doublesex*, sex determination, sexual differentiation, sex reversal, parental RNAi, Hymenoptera

## Abstract

**Simple Summary:**

The sawfly, *Athalia rosae*, exploits a haplodiploid mode of reproduction, in which fertilized eggs develop into diploid females, whereas unfertilized eggs parthenogenetically develop into haploid males. The *doublesex* (*dsx*) gene is a well-conserved transcription factor that regulates sexual differentiation in insects. In the present study, we knocked down the *A. rosae* ortholog of *dsx* (*Ardsx*) during several developmental stages with repeated double-stranded RNA (dsRNA) injections. As a result, knockdown of *Ardsx* in haploid males caused almost complete male-to-female sex reversal, but the resulting eggs were infertile. The same knockdown approach using diploid males caused complete male-to-female sex reversal; they were able to produce fertile eggs and exhibited female behaviors. The same RNAi treatment did not affect female differentiation. These results demonstrated that *dsx* in the sawfly is essential for male development and its depletion caused complete male-to-female sex reversal. This is the first demonstration of functional depletion of *dsx* not causing intersexuality but inducing total sex reversal in males instead.

**Abstract:**

The *doublesex* (*dsx*) gene, which encodes a transcription factor, regulates sexual differentiation in insects. Sex-specific splicing of *dsx* occurs to yield male- and female-specific isoforms, which promote male and female development, respectively. Thus, functional disruption of *dsx* leads to an intersexual phenotype in both sexes. We previously identified a *dsx* ortholog in the sawfly, *Athalia rosae*. Similar to *dsx* in other insects, *dsx* in the sawfly yields different isoforms in males and females as a result of alternative splicing. The sawfly exploits a haplodiploid mode of reproduction, in which fertilized eggs develop into diploid females, whereas unfertilized eggs parthenogenetically develop into haploid males. In the present study, we knocked down the *A. rosae* ortholog of *dsx* (*Ardsx*) during several developmental stages with repeated double-stranded RNA (dsRNA) injections. Knockdown of *Ardsx* via parental RNA interference (RNAi), which enables knockdown of genes in offspring embryos, led to a lack of internal and external genitalia in haploid male progeny. Additional injection of dsRNA targeting *Ardsx* in these animals caused almost complete male-to-female sex reversal, but the resulting eggs were infertile. Notably, the same knockdown approach using diploid males obtained by sib-crossing caused complete male-to-female sex reversal; they were morphologically and behaviorally females. The same RNAi treatment did not affect female differentiation. These results indicate that *dsx* in the sawfly is essential for male development and its depletion caused complete male-to-female sex reversal. This is the first demonstration of functional depletion of *dsx* not causing intersexuality but inducing total sex reversal in males instead.

## 1. Introduction

The sex-determination mechanism, which decides an individual’s sexual fate, and the sexual-differentiation mechanism, which governs sexually dimorphic traits, are both essential for sexually reproducing organisms. Sex determination and sexual differentiation are controlled by a genetic cascade consisting of several sex-determining genes. Only the final element of the cascade is well-conserved, whereas upstream genes are highly evolutionary labile [1]. Studies suggest that *doublesex* (*dsx*) acts at a downstream position in the sex-determination cascade and regulates sexual differentiation in a wide variety of insect species [2,3]. The *dsx* gene has also been identified in five cladoceran species, including *Daphnia magna* and *D. pulex*, which are the closest relatives to insects [4,5].

In *Drosophila melanogaster*, the primary *dsx* transcript undergoes sex-specific alternative splicing to produce a male-specific (DSXM) or a female-specific (DSXF) protein isoform [6]. The DSXF and DSXM proteins regulate the sex-specific transcription of target genes to develop the female or male body phenotype [7,8,9]. Unlike in insects, *dsx* in *D. magna* is expressed predominantly in males and is only required for male development [4,5].

As the name implies, depleting *dsx* causes defects in male and female development, resulting in intersexuality in both sexes. Functional disruption of *dsx* causes the intersexual phenotype in several insect species. In the fruit fly *Drosophila melanogaster*, *dsx*-mutant females exhibit a male-specific pattern of abdominal pigmentation and develop a sex comb, which is a male-specific organ, whereas male-specific abdominal pigmentation and sex-comb formation are repressed in *dsx*-mutant males [10]. The *dsx*-mutant female silkworm, *Bombyx mori*, has a clasper-like structure and eight abdominal segments, both of which are male-specific traits, whereas the *dsx*-mutant male develops female-specific genital papillae [11,12]. In the Japanese horned beetle *Trypoxylus dichotomus*, which is famous for its large horns (on the head and thorax) in males, RNA interference (RNAi)-mediated *dsx* knockdown causes atrophy of the head horn and loss of the thoracic horn. On the other hand, *dsx* knockdown in females results in the appearance of a head horn [13]. However, the sex reversal effects caused by the cases of *dsx* dysfunction characterized so far are all partial; thus, functional disruption of *dsx* leads to an intersexual phenotype in most cases [14,15,16,17,18,19,20]. These results may be inevitable because *dsx* produces male- and female-specific isoforms, each of which is involved in male and female development.

The sawfly, *Athalia rosae* (Hymenoptera: Tenthredinidae), belongs to the Symphyta, which is the most primitive infraorder in the Hymenoptera. Similar to other hymenopteran insects, the sexual fate of the sawfly is determined by the single-locus complementary sex determination (CSD) system, in which heterozygosity at a single locus (the CSD locus) determines femaleness in diploid individuals, whereas haploid individuals are hemizygous for the CSD locus and thus develop into males [21,22]. In a previous study, we successfully identified a *dsx* ortholog in the sawfly and designated the gene *Ardsx* [23]. *Ardsx* pre-mRNA was spliced alternatively in a sex-dependent manner, yielding female- and male-specific isoforms (designated as *ArdsxF* and *ArdsxM*, respectively) at almost all developmental stages. Our previous findings demonstrate that *ArdsxM* is required for male development [23]. Thus, like *dsx* orthologs so far identified in other holometabolous insects, *Ardsx* is likely to act at the bottom of the sex determination cascade and regulate sexual development in the sawfly. Notably, unlike other holometabolous insects, knockdown of *Ardsx* during the pupal stage in males caused almost complete male-to-female sex reversal in the external genitalia, whereas the same knockdown in females had no effect on sexual phenotype. However, the effect of *Ardsx* knockdown on male differentiation of internal genitalia, including the gonads, was extremely weak. This may be because the effect of *Ardsx* knockdown is transient—*Ardsx* expression was transiently knocked down from the last-instar larval stage to the adult stage in the previous study. It remains unclear whether *Ardsx* knockdown causes total male-to-female transformation and is dispensable for female development.

To address this question, it is important to knock down *Ardsx* expression during the entire lifespan, including the embryonic stage when *Ardsx* initiates sex-dimorphic expression [23]. In the sawfly, injecting double-stranded RNA (dsRNA) into the parental females (parental RNAi) is more efficient than embryonic injection in terms of penetrance of the effect [24]. In the present study, repeated injections of dsRNAs targeting *Ardsx* were given to females after parental RNAi in order to extend the duration of RNAi-mediated knockdown of *Ardsx* expression for as long as possible. This RNAi treatment caused male-to-female sex reversal, not only in the external genitalia, but also in the gonads: the RNAi-treated males had ovaries containing fertile eggs. On the other hand, females that received the same RNAi treatment exhibited a normal phenotype, supporting our previous finding that *Ardsx* is dispensable for female differentiation. Thus, unlike in other insect species, functional depletion of *dsx* in the sawfly did not cause an intersexual phenotype but induced total male-to-female transformation. Some evolutionary implications of these findings will be discussed later.

## 2. Materials and Methods

### 2.1. Insects

Wild-type sawflies (*A. rosae*) were reared continuously at 25 °C under 16-h-light/8-h-dark conditions and fed fresh Japanese radish leaves (Sakata Seed Corporation, Yokohama, Japan) cultivated as described previously [23]. Diploid females were acquired from fertilized eggs laid in radish leaves. Haploid males were obtained by parthenogenetic activation of mature unfertilized eggs according to a previously described protocol [25]. The eggs were stored in plastic containers (Sanplatec, Tokyo, Japan) with sufficient humidity (>90% relative humidity). A *yellow fat body* (*yfb*) mutant strain was also used in the present study. The wild-type fat body is greenish blue in color but animals homozygous for *yfb* have fat bodies that are yellow [25]. Individuals heterozygous for *yfb* display intermediate coloration (Appendix A). Based on the phenotype, we discriminated diploid males in fertilized eggs from haploid males in unfertilized eggs. To obtain diploid males, a single diploid female homozygous for *yfb* (*yfb*/*yfb*) was mated with a wild-type haploid male, and their progeny were subjected to sib-crossing, where a single diploid female (+/*yfb*) was crossed with a single haploid male (+ or *yfb*). Such interbreeding was repeated multiple times. The resulting males heterozygous for *yfb* (+/*yfb*) were treated as diploids. To discriminate diploid males from females, molecular sexing of each animal was determined according to the expression pattern of *Ardsx*, investigated by RT-PCR, as described below, at more than 7 days after emergence, when the effect of *Ardsx* knockdown had disappeared. Individuals that preferentially expressed female-type *Ardsx* (*ArdsxF*) were recognized as females, whereas individuals that predominantly expressed male-type *Ardsx* (*ArdsxM*) were defined as males (Appendix A).

### 2.2. RNA Extraction and RT-PCR

Extraction of total RNA was performed using ISOGEN (Nippon Gene, Tokyo, Japan), according to a previously described protocol [23]. The RT-PCR analyses were performed as described previously [26]. The ArEF1-LP and ArEF1-RP primers were used to amplify *A. rosae* elongation factor-1 alpha (*EF-1 alpha*) as an internal standard for the RT-PCR [23]. The primer sequences used in this study are shown in Appendix A. The PCR products were analyzed using 2% agarose gel electrophoresis and visualized with ethidium bromide (Bio-Rad, Hercules, CA, USA).

### 2.3. Quantitative Real-Time RT-PCR

Quantitative RT-PCR assays were performed according to a previously described protocol [26]. The primers used to amplify *Ardsx* (Ardsx Real Time F1 and Ardsx Real Time R1) were the same as those used in our previous study [23]. The ArEF1-LP and ArEF1-RP primers were used to amplify the *EF-1 alpha* gene as an internal standard. The threshold cycle (CT) value was normalized with the CT value of the *EF-1 alpha* gene using Multiple RQ software (TaKaRa Bio Inc., Kyoto, Japan). We confirmed that the above primer sets do not produce off-target products by checking dissociation curves of the qPCR products. The relative value of *Ardsx* expression against *EF-1 alpha* expression was obtained in quadruplicate and the ratio of *Ardsx* expression calculated.

### 2.4. Preparation of dsRNAs

Two sequences shared among the male and female *Ardsx* isoforms were amplified with the primer pairs ArdsxdsRNAF1-ArdsxdsRNAR1 and ArdsxdsRNAF2-ArdsxdsRNAR2, as described previously [23], and served as a DNA template for dsRNA synthesis. Each primer contained a T7 promoter site. The dsRNA synthesis was performed according to a previously described protocol [26].

### 2.5. Injection of dsRNA into Insects

The dsRNAs were injected into larvae and pupae according to a previously described protocol [23]. Parental RNAi was applied as described previously [24]. Female pupae at three days after pupation were subjected to parental RNAi. dsRNAs were injected at the suture between the third and fourth abdominal segments of a pupa. Mature unfertilized eggs collected from some of the parental RNAi treated females were parthenogenetically activated to obtain haploid males according to a previously described protocol [25]. To achieve knockdown of *Ardsx* expression in order to prolong the duration of RNAi-mediated mRNA knockdown, further injections of dsRNA into progeny obtained via parental RNAi were performed at the second, third, and final instar larval stages. When the larvae were subjected to RNAi, dsRNAs were injected into the dorsal hemocoel in the second abdominal segment of a larvae. Approximately 1 μL of dsRNA solution with a concentration of 100 ng dsRNA/μL was injected into each animal.

### 2.6. Observation of Internal Reproductive and Genital Organs

Cuticle specimens of external genitalia were prepared as described previously [23]. The specimens were observed under a stereoscope (SZX 7, OLYMPUS, Tokyo, Japan). A CCD camera (SP 7, OLYMPUS, Tokyo, Japan) mounted on the stereoscope was used to capture images. The images were analyzed with CellSens standard software (OLYMPUS, Tokyo, Japan). Adult gonads and internal reproductive organs were dissected out in 1× phosphate-buffered saline immediately after emergence. The dissected organs were observed using the stereoscope as described above.

### 2.7. Behavioral Assays

All behavioral experiments were carried out near a window with sunlight at 25 °C. Males and virgin females were typically used at 7 days post-eclosion. *Ardsx*-knockdown males were used at least 14 days after eclosion, as we discovered that egg maturation in *Ardsx*-knockdown males took longer than in wild-type females. Naïve males, which had never met a female and had never experienced mating, and virgin females were isolated in vials before use. To examine whether *Ardsx*-knockdown males exhibited courtship behaviors, two wild-type virgin females were introduced into a vial (30 mm in diameter, 72 mm in height) with two *Ardsx*-knockdown males. To investigate whether *Ardsx*-knockdown males were able to attract naïve wild-type males, a single *Ardsx*-knockdown male was introduced into a vial with a naïve wild-type male. Video recording (iPhone 7, Apple, Cupertino, CA, USA) started immediately after introducing the virgin females (or pseudofemales). The typical courtship behavior of the sawfly based on previous reports [27,28,29] is as follows: (1) The male sawfly explores the female in a waiting position. (2) When the male visually recognizes the female, the male chases the female, and touches and licks the female’s body surface. (3) The male then uses the tail clasper to grab the female’s copulatory organ and insert his penis (defined as copulation). (4) The male and the female stay still for a few to approximately 30 min in a tail-to-tail connection (defined as copulation success). (5) Once the male has completed spermatophore transfer to the female, the male leaves the female. The following three courtship parameters were analyzed. *Courtship index*: the fraction of time spent in any aspect of courtship behavior, including chasing, touching, licking, and attempted copulation within the first 10 min of the assay. *Copulation latency*: the time from presentation to copulation. *Copulation success*: percentage of males achieving copulation within the first hour of the assay.

### 2.8. Statictics

All statistical processing was performed using Easy R (EZR) software (https://www.jichi.ac.jp/saitama-sct/SaitamaHP.files/download.html, accessed on 6 September 2021). The Shapiro–Wilk test was used to evaluate whether the data obtained by our experiments showed a normal distribution. Since the number of samples was less than 25 and did not show a normal distribution, the significant difference test between the two groups was performed using the Mann–Whitney U test. The data obtained in this study were displayed in box plot format. The box plot was created using EZR software with default settings.

## 3. Results

### 3.1. Effects of Repeated Injections of Ardsx dsRNAs on Ardsx Expression

In a previous study, we investigated the role of *Ardsx* on sexual development in the sawfly by transiently knocking down *Ardsx* expression during the early pupal stage [23]. However, to more precisely understand the function of *Ardsx* during sexual differentiation, it is ideal to knock down *Ardsx* expression during the entire lifespan, including at the embryonic stage when *Ardsx* initiates sex-dimorphic expression. In the sawfly, injecting dsRNA into parental females (parental RNAi) is more efficient than embryonic injection in terms of penetrance of the effect [24]. In the present study, repeated injections of dsRNAs targeting *Ardsx* were given to females produced after parental RNAi in order to extend duration of RNAi-mediated knockdown of *Ardsx* expression for as long as possible (Figure 1A,B).

Parental injection of dsRNAs targeting a region common between the male and female *Ardsx* isoforms (Figure 1A) caused a significant reduction in *Ardsx* mRNA levels in male and female progeny at the embryonic stage (Figure 1C, left panel). Further injections of dsRNA into progeny obtained via parental RNAi, at the second, third, and final instar larval stages (Figure 1B), significantly decreased the expression level of *Ardsx* in the third instar and day-3 pupal stages (Figure 1C, middle and right panels).

These results suggest that the repeated injections of dsRNA targeting *Ardsx* given to animals produced after parental RNAi was performed effectively repressed *Ardsx* expression levels during several developmental stages, including embryonic, larval, and pupal stages.

### 3.2. Effects of Knockdown of Ardsx Expression on Sexual Differentiation of Genitalia

Next, we investigated the effects of repeated injections of Ardsx1 dsRNA on the sexual development of females and haploid males. No morphological changes were observed in either the external or internal genitalia in *Ardsx*-knockdown females obtained via repeated injections of Ardsx1 dsRNA (compare Figure 2C,F,I; also compare Figure 3C,F,I). As shown in Table 1, all the examined *Ardsx*-knockdown females showed the normal sexual phenotype (Table 1). On the other hand, the external genitalia in *Ardsx*-knockdown haploid males displayed complete male-to-female sex reversal, resulting in the formation of female-specific organs, such as the oviduct and sheath (compare Figure 2D,G,J). In addition, a morphological analysis of the internal genitalia revealed that these *Ardsx*-knockdown haploid males did not develop testes but instead formed ovaries containing mature-looking eggs (compare Figure 3D,G,J). These male-to-female sex reversals in external genitalia and gonads were observed in most of the examined haploid males (Table 1). A similar male-to-female sex reversal phenotype was also observed in the internal genitalia of haploid males obtained by repeated injections of another dsRNA (Ardsx2) targeting *Ardsx* mRNA (Appendix A).

### 3.3. Effects of Knockdown of Ardsx Expression on Oogenesis

Next, we investigated the effect of *Ardsx* knockdown on oogenesis. Ovarian eggs produced by *Ardsx*-knockdown females obtained via repeated injections of Ardsx1 dsRNA looked normal in appearance (compare Figure 4A–C). There was no significant difference in the mean number of eggs obtained from a single female between wild-type and *Ardsx*-knockdown females (Figure 4F). Parthenogenetic activation of ovarian eggs demonstrated that eggs produced by *Ardsx*-knockdown females had a normal fertility (Figure 4G).

Eggs obtained from ovaries of *Ardsx*-knockdown haploid males seemed normal in appearance (Figure 4D). However, the mean number of eggs in *Ardsx*-knockdown haploid males was significantly lower than that in wild-type females (Figure 4F). No hatched larvae were obtained from these eggs even after parthenogenetic activation (Figure 4G).

Reductional maturation division (meiosis) does not occur during spermatogenesis in the male sawfly [30] because males are generally haploid; thus, meiosis is not needed to produce sperm. By contrast, the maturation of eggs produced by diploid females requires a meiotic process. It is postulated that the male-to-female reversal observed in *Ardsx*- knockdown males caused meiosis to occur but that the meiosis might not have proceeded properly because the cells are all haploid.

To verify whether this hypothesis was correct, diploid males were subjected to the same RNAi experiment (see Section 2). As with haploid males, *Ardsx* knockdown of diploid males caused male-to-female sex reversal as observed in both the external genitalia and gonads (compare Figure 2E,H,K; also compare Figure 3E,H,K). The extent of sex reversal was more severe than that observed in haploid males (compare Figure 2J,K; also compare Figure 3J,K). As shown in Table 1, these male-to-female sex reversals in external genitalia and gonads were observed in most of the examined diploid males (Table 1). These males developed ovaries carrying mature eggs (Figure 4E). The internal genitalia of diploid males obtained by repeated injections of another dsRNA (Ardsx2) targeting *Ardsx* mRNA also showed the same sex reversal phenotype (Appendix A). Although the number of mature eggs produced by the *Ardsx*-knockdown diploid males was significantly lower than that by wild-type females (Figure 4F), hatchability of these eggs was almost the same as that of wild-type females (Figure 4G). These results support the validity of the hypothesis that ploidy may affect the normal progression of meiosis and subsequent egg maturation. Based on these results, since *Ardsx*-knockdown diploid males showed complete female-to-male sex reversal in the external genitalia and gonads, and produced eggs with normal fertility, we designated them as pseudofemales for convenience.

### 3.4. Effects of Ardsx Knockdown on Sexual Behavior

To investigate the effects of *Ardsx* knockdown on sexual behavior, we evaluated whether the *Ardsx*-knockdown males copulated with wild-type virgin females. As shown in Figure 5, no significant difference in copulation latency, which is the time it takes for a male to initiate copulation with a virgin female, was observed between control males and *Ardsx*-knockdown haploid males and pseudofemales (=*Ardsx*-knockdown diploid males) (also see Appendix A). However, the courtship index, which is the fraction of time spent in any aspect of courtship during the first 10 min of the assay, was significantly lower in both the *Ardsx*-knockdown haploid males and pseudofemales than in control males (Figure 5B). These results suggest that knockdown of *Ardsx* expression had no qualitative effect on male courtship behavior but caused a significant decrease in its activity. All *Ardsx*-knockdown haploid males and pseudofemales failed to copulate with females (Figure 5C). This could be attributed simply to the complete loss of male external genitalia in *Ardsx*-knockdown males, as shown in Figure 2J,K.

To evaluate the possibility that the sex-reversed males might exhibit female behavior, we performed the reciprocal experiment: *Ardsx*-knockdown haploid males and pseudofemales were put together with wild-type naïve males, which had never met a female and had never experienced mating. As a result, only pseudofemales attracted and fully activated courtship behavior by wild-type naïve males (Figure 5D,E). Moreover, these pseudofemales successfully copulated with males (Figure 5F and Appendix A). These results demonstrated that knockdown of *Ardsx* not only induced morphological femaleness but also caused behavioral male-to-female sex reversal in diploid males.

## 4. Discussion

This study supported our previous report that *Ardsx* is not required for female development in the sawfly [23]. A caveat is that we were not able to confirm whether the protein level of ArDSX was reduced due to the lack of antibodies that specifically recognize the ArDSX protein. Therefore, the absence of abnormalities in female development in *Ardsx* RNAi treated females may simply reflect an insufficient level of knockdown of ArDSX protein expression in females. However, it is unlikely that ArDSX protein level was not reduced only in females, since the extent of reduction of *Ardsx* mRNA by our RNAi was almost similar in both males and females.

The *dsx* gene is one of the doublesex/mab-3-related genes (Dmrt genes). In insects, *dsx* is characteristic in that it produces different isoforms in males and females as a result of alternative splicing, and each isoform is responsible for male and female sexual development [16,18,20,31,32,33,34,35,36]. In cladoceran species such as *Daphnia magna* and *D*. *pulex*, *dsx* orthologs are essential for male development and do not exhibit sex-specific splicing (Figure 6) [4,5]. In vertebrates, orthologs of *DMRT1* are predominantly expressed in males and are only required to guide male sex determination [37]. Ledoń-Rettig et al. postulated that non-insect arthropods may exhibit a putative ancestral condition of *dsx*, in which *dsx* does not undergo sex-specific splicing and is male-specifically expressed to promote male development [38]. They also hypothesized that sex-specific splicing of *dsx* in insects is derived from the non-spliced ancestral condition, and thus, the female DSX isoform is relatively novel and does not contribute largely to female development in several tissues [38]. In accordance with their idea, *dsx* in the German cockroach, *Blattella germanica*, which belongs to the most basal insect order studied about *dsx* to date, undergoes sex-specific splicing to yield female and male *dsx* isoforms, but this process is only required for male sexual differentiation (Figure 6) [39]. Hymenoptera is the most basal lineage in the phylogeny of holometabolous insects (Superorder Endopterygota) [40,41,42,43]. Among the hymenopteran species, the sawfly, *A*. *rosae* (Hymenoptera: Tenthredinidae), belongs to the Symphyta infraorder, which is the most primitive infraorder in the Hymenoptera. Thus, it is quite reasonable to speculate that *dsx* in the sawfly is only required for male development and is dispensable for female development (Figure 6). However, at the present time, we cannot rule out the possibility that the role of *dsx* in female development may simply be lost secondarily in the sawfly. Further detailed analysis of the function of *dsx* in the wide variety of species that belong to the Symphyta infraorder and other hymenopteran species will be required to clarify this point.

The most remarkable finding in the present study is that the knockdown of *Ardsx* in diploid males yielded mature eggs with normal fertility (Figure 4E,G). In the vinegar fly *Drosophila melanogaster*, sexual differentiation in germ cells is controlled by a sex-determining cascade that is unique to germ cells. In female germ cells (XX), oogenesis occurs even if the sex of the surrounding somatic cells is male. DSX is dispensable for oogenesis, and factors that are expressed only in XX germ cells, such as a protein product of *ovarian tumor* (*otu*), as well as the germline-specific isoforms of OVO and SXL, are essential for oogenesis [44,45,46]. Oogenesis in the sawfly is similar to that in *D*. *melanogaster*, in that DSX is not required for oogenesis, because *Ardsx*-knockdown individuals produce normal eggs (Figure 4E,G). However, unlike *Drosophila*, oogenesis in the sawfly proceeds normally even if the genetic sex of the germ cells was male. Sexual differentiation in the germ cells of the sawfly may be regulated in the same way as in somatic cells. This may be related to the fact that the sawfly does not have sex chromosomes. Further research on germ-cell sexual differentiation in insects without sex chromosomes is needed to examine this hypothesis.

As described above, the diploid males in whom *Ardsx* expression was knocked down behaved as if they were female. They accepted copulation from males (Figure 5D–F and Appendix A). These results strongly suggest that *Ardsx* is dispensable for female differentiation of neural networks, as is the case for reproductive organs and germ cells. Expression of *ArdsxM* in wild-type males would repress the development of female-specific neural networks.

Despite *Ardsx* knockdown frum in males, the *Ardsx*-knockdown males still exhibited courtship behavior (Figure 5A and Appendix A). In *D*. *melanogaster*, *fruitless* (*fru*) plays a pivotal role in the development of the male-specific nervous system that is required for male sexual behavior [47,48,49]. Mutations in male-specific *fru* transcripts (*fru^M^*) cause defects in male courtship behavior and sexual identity [44,50,51,52,53,54]. Recent insights from non-drosophilid insects, such as the housefly *Musca domestica*, the mosquitoes *Anopheles gambieae* and *Aedes aegypti*, the parasitic wasp *Nasonia vitripennis*, and several hemimetabolous insects, including the desert locust *Schistocerca gregaria* and the German cockroach *B. germanica*, suggest a conserved evolutionary role of the Fru protein in male courtship behavior [55,56,57,58,59,60]. Taken together with these findings, it is reasonable to speculate that *fru*, not *dsx*, plays an essential role in the development of male-specific neural networks in the sawfly.

However, in *A. rosae*, the courtship index for *Ardsx*-knockdown males was significantly lower than that for wild-type males (Figure 5B). This is reminiscent of the general decrease in the level of male courtship displayed by *dsx*-mutant males of *D*. *melanogaster* [61,62]. In *Drosophila*, the male-specific isoform of DSX (DSXM) is not necessary for most aspects of male courtship behavior, but it is required for the male courtship song [60,61]. DSXM is necessary and sufficient for the acquisition of the potential for experience-dependent male courtship behavior [63]. Several subsets of neurons in the male-specific central nervous system require *fru^M^* and *dsxM* functions to accomplish normal male development [64]. Also, *dsxM* acts together with *fru^M^* in the differentiation of male-specific neurons in the abdominal ganglion [64]. Similar to *dsx* in *Drosophila*, *Ardsx* has a partial but important role in specifying all aspects of male courtship behavior in the sawfly.

Interestingly, when haploid males were subjected to repeated injections of Ardsx1 dsRNA, the resulting *Ardsx*-knockdown males did not attract wild-type males (Figure 5D–F). These results suggest that diploidy is not only essential for producing fertile eggs but also important for producing normal female behaviors. This is most likely due to the incomplete male-to-female sex reversal observed in the *Ardsx*-knockdown haploid males (Figure 2J and Figure 3J). How does diploidy contribute to complete male-to-female sex reversal? A partial explanation may be found in the sex-determining mechanism observed in the parasitic wasp *Nasonia*, which has no *csd* locus but employs a haplodiploid sex-determination system [65,66]. In *Nasonia*, expression of the feminizing factor *Nasonia vitripennis transformer* (*Nvtra*) occurs only in fertilized eggs, where maternal and paternal genomes coexist [67]. Unfertilized eggs, which only have the maternal genome, develop as males because transcription of the maternally derived *Nvtra* allele is prevented by maternal imprinting [64]. It is possible to speculate that the expression of some genes pivotal to feminization of the sawfly are modified by genomic imprinting, resulting in normal female development only in individuals that have both paternal and maternal genomes. Further studies are needed to validate this hypothesis.

**Figure 6 insects-12-00849-f006:**
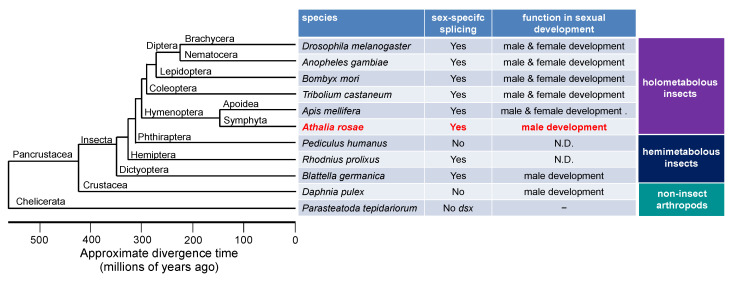
Summary of the status of sex-specific splicing of *dsx* and its function in sex differentiation in some representative species of insects and non-insect arthropods. The phylogenetic tree was drawn with reference to Figure 1 of a previous study published by the Honeybee Genome Sequencing Consortium (2006) [68]. “Yes” means that *dsx* pre-mRNA undergoes sex-specific splicing to yield male- and female-specific isoforms. “No” indicates that *dsx* pre-mRNA is not sex-specifically spliced. “No *dsx*” indicates that there is no *dsx* ortholog in the genome. N.D., not determined.

## 5. Conclusions

In *D. melanogaster*, the *dsx* gene acts at the bottom of the sex-determination cascade that induces appropriate sexual differentiation in each sex according to upstream genetic sex-determining signals [2,3]. Thus, the absence of *dsx* leads to the intersexual phenotype in both sexes [14,15,16,17,18,19,20]. This strongly suggests that the default mode of sexual differentiation in insects is intersex. The only exception has been reported in the German cockroach, *B. germanica*, where knockdown of *dsx* causes partial male-to-female sex reversal but has no effect on female development [39]. Distinct from the case in the German cockroach, we demonstrated here that knockdown of *dsx* by repeated injections of dsRNA in the sawfly induced complete male-to-female transformation. The resulting pseudofemales successfully copulated with wild-type males and produced fertile eggs. Remarkably, the total male-to-female transformation was observed only when diploid males were subjected to the RNAi treatment. A caveat is that we were not able to confirm whether *Ardsx* expression in the examined diploid males was indeed knocked down during development from the embryonic to pupal stages by the RNAi, as their genetic sex can only be identified by RT-PCR analysis performed at least seven days after emergence (see Section 2). However, since the expression level of *Ardsx* was significantly reduced in diploid females (Figure 1C), we expected that the same RNAi treatment would efficiently decrease *Ardsx* expression in diploid males as well. These results suggest that diploidy is also important for normal female development. The sexual fate of the sawfly is determined by the CSD system, in which heterozygosity at the CSD locus determines femaleness in diploid individuals, whereas haploid or diploid individuals hemizygous or homozygous for the CSD locus develop into males [21,22]. At first glance, in the CSD system, heterozygosity at the CSD locus alone seems to be sufficient for female development. However, our study suggested that diploidy is also important for femaleness in hymenopteran species whose sexual fate is determined by the CSD system. We hypothesized that the expression of some genes pivotal to feminization of the sawfly are modified by genomic imprinting, resulting in normal female development only in individuals that have both paternal and maternal genomes. Alternatively, some part of the female differentiation may simply require two copies of effector genes for the induction of proper female development. It is considered that, in diploid males, the expression of such genes required for femaleness is repressed by the action of male-type *Ardsx*; thus, they do not exhibit femaleness even though they have diploid genomes. Further studies are needed to validate this hypothesis.

## Figures and Tables

**Figure 1 insects-12-00849-f001:**
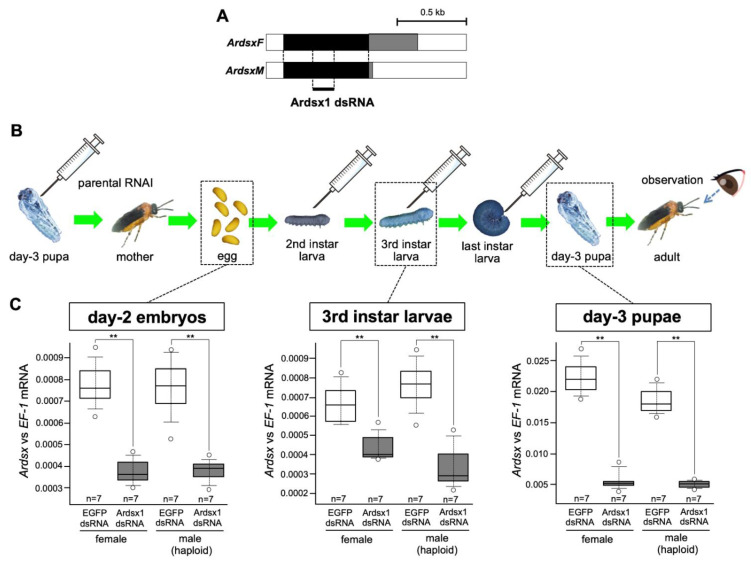
Outline of the RNAi methods used in this study. (**A**) Schematic diagram of the *ArdsxF* and *ArdsxM* mRNAs. The white regions indicate untranslated regions (UTRs). The black regions represent the open-reading frames (ORFs) shared between the *ArdsxF* and *ArdsM* mRNAs. The gray regions indicate mRNA sequence-encoding sex-specific ORFs. Positions of the dsRNAs targeting *Ardsx* used in this study are indicated by bold lines. (**B**) Schematic diagram of when and how many times Ardsx1 dsRNA was injected during development and the expected duration of *Ardsx*-expression knockdown after each round of RNAi. The syringe illustration indicates the timing of the dsRNA injection. (**C**) Effect of the repeated injections of Ardsx1 dsRNA on *Ardsx* expression. Left panel: The expression level of *Ardsx* mRNA in the day-2 embryos (three days before hatching), obtained from females injected with dsRNA targeting *Ardsx*, were determined using quantitative RT-PCR (qRT-PCR). Middle panel: Additional injections of dsRNA targeting *Ardsx* were performed at the second and the third instar larval stages and the expression level of *Ardsx* mRNA in the third instar larvae was quantified by qRT-PCR. Right panel: The dsRNA was further injected at the final instar larval stage and the expression level of *Ardsx* mRNA in the day-3 pupae was determined by the same qRT-PCR. *EF-1alpha* served as an internal standard (see Section 2). The values shown by each sample and their distributions were represented by box-and-whisker plot. ** Significant difference at the 0.02 level (Mann–Whitney U test) compared with the negative control group, where EGFP dsRNA was injected into animals in the same way.

**Figure 2 insects-12-00849-f002:**
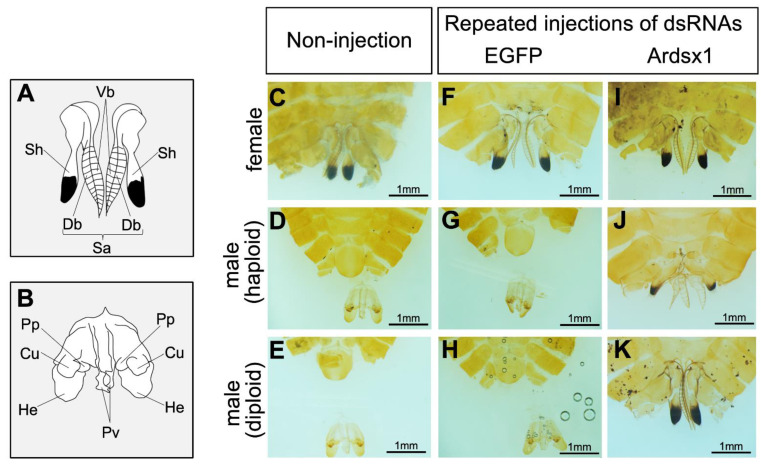
Effects of RNAi-mediated knockdown of *Ardsx* on sexual development of external genital organs. (**A**,**B**) Schematic diagrams of female (**A**) and male (**B**) external genital organs. Db, dorsal pair of blades; Sa, saw; Sh, sheath; Vb, ventral pair of blades; Cu, cuspis; He, herpe; Pp, parapenis; Pv, penis valve. (**C**–**K**) Ventral view of the external genitalia at the adult stage. Female (**C**), haploid male (**D**), and diploid male (**E**) without injection of dsRNA. Negative control female (**F**), haploid male (**G**), and diploid male (**H**) obtained via repeated injections of EGFP dsRNA. *Ardsx*-knockdown female (**I**), haploid male (**J**), and diploid male (**K**) obtained via repeated injections of Ardsx1 dsRNA.

**Figure 3 insects-12-00849-f003:**
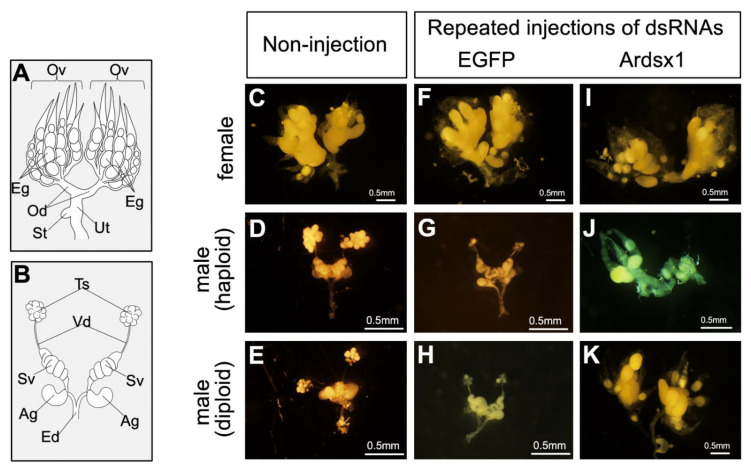
Effects of RNAi-mediated knockdown of *Ardsx* on sexual development of internal genital organs and oogenesis. (**A**,**B**) Schematic diagrams of female (**A**) and male (**B**) internal reproductive organs. Eg, egg; Ov, ovary; Od, oviduct; St, spermatheca; Ut, uterus; Ag, accessory gland; Ed, ejaculatory duct; Sv, seminal vesicle; Ts, testis; Vd, vas deferens. (**C**–**K**) Ventral view of the internal genitalia at the adult stage. Female (**C**), haploid male, and diploid male (**E**) without injection of dsRNA. Negative control female (**F**), haploid male (**G**), and diploid male (**H**) obtained via repeated injections of EGFP dsRNA. *Ardsx*-knockdown female (**I**), haploid male (**J**), and diploid male (**K**) obtained via repeated injections of Ardsx1 dsRNA.

**Figure 4 insects-12-00849-f004:**
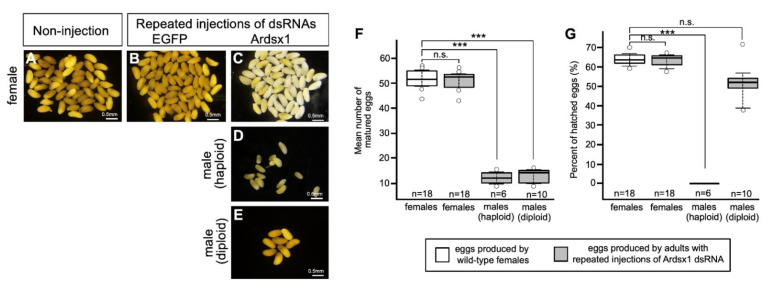
Effects of RNAi-mediated knockdown of *Ardsx* on oogenesis. Mature eggs obtained from ovaries in wild-type female (**A**), negative control female (**B**), *Ardsx*-knockdown female (**C**), *Ardsx*-knockdown haploid male (**D**), and *Ardsx*-knockdown diploid male (**E**). (**F**) The number of eggs obtained from wild-type females and *Ardsx*-knockdown haploid and diploid males produced via repeated injections of Ardsx dsRNA. (**G**) Hatchability of parthenogenetically activated eggs produced by wild-type females and *Ardsx*-knockdown haploid and diploid males produced via repeated injections of Ardsx1 dsRNA. The values shown by each sample and their distributions were represented by box-and-whisker plot. *** Significant difference at the 0.01 level (Mann–Whitney U test) compared with the wild-type females. n.s. means no significant difference between two groups (Mann–Whitney U test).

**Figure 5 insects-12-00849-f005:**
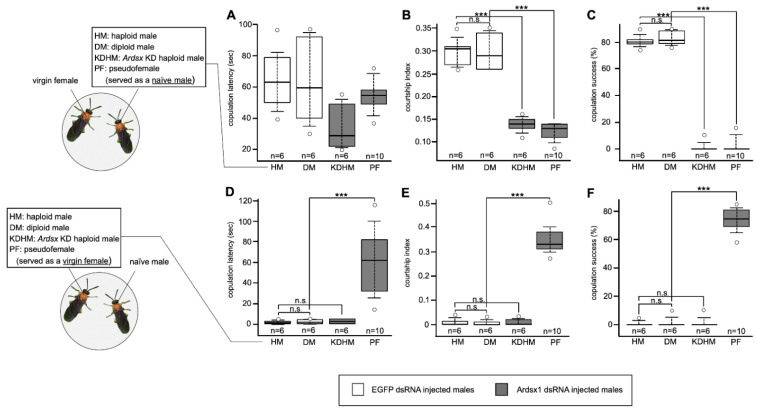
Effects of *Ardsx* knockdown on sexual behavior. (**A**–**C**) To evaluate whether *Ardsx*-knockdown males copulated with wild-type virgin females, either an *Ardsx*-knockdown haploid male or a pseudofemale (=*Ardsx*-knockdown diploid male) was placed with a virgin female. (**A**) Copulation latency, the time it takes for a male to initiate copulation with a virgin female; (**B**) the courtship index, the fraction of time spent in any aspect of courtship during the first 10 min of the assay; and (**C**) copulation success, the percentage of pairs that copulated within the first 1 h, were compared among different types of males. (**D**,**E**) To examine whether wild-type naïve males were attracted by *Ardsx*-knockdown haploid males or pseudofemales and copulated with them, a naïve male was put together either with an *Ardsx*-knockdown haploid male or a pseudofemale, and (**D**) copulation latency, (**E**) the courtship index, and (**F**) copulation success were recorded. The values shown by each sample and their distributions were represented by box-and-whisker plot. *** Significant difference at the 0.01 level (Mann–Whitney U test) compared with the negative control haploid males. n.s means no significant difference between two groups (Mann–Whitney U test).

**Table 1 insects-12-00849-t001:** Number of injected animals, adults subjected to the analysis, and adults that exhibited sex-reversal phenotype.

dsRNA	Sexuality	No. of Injected Animals			No. of Investigated Adults	Sexual Phenotypes *^2^	
		2nd Instar	3rd Instar	Last Instar		Normal	Sex-Reversal
EGFP	female	50	26	19	12	12	0
	male (haploid)	63	27	20	13	13	0
	male (diploid)	N/A *^1^	N/A	N/A	7	7	0
Ardsx1	female	59	35	26	18	18	0
	male (haploid)	54	29	20	13	1	12
	male (diploid)	N/A	N/A	N/A	12	1	11

*^1^: N/A means not acquired. We were not able to distinguish diploid males from diploid females during the larval stages because their genetic sex can only be identified by RT-PCR analysis performed at least seven days after emergence, when all the experiments were completed (see Section 2). *^2^: Sexual phenotype includes morphologies of external and internal genital organs.

## Data Availability

The mutant line (*yfb*) used in this study is continuously reared and passaged at the Laboratory of Bio-resource Regulation, Department of Integrated Biosciences, Graduate School of Frontier Sciences, The University of Tokyo and Division of Applied Genetics, Institute of Agrobiological Sciences, NARO. All data obtained or analyzed during the present study are available from the corresponding author upon reasonable request.

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
