# Peer review of "Effects of Functional Depletion of Doublesex on Male Development in the Sawfly, Athalia rosae"

_insects, 2021, doi:10.3390/insects12100849_

Round 1

Reviewer 1 Report

The manuscript “Effects of functional depletion of doublesex on male development in the sawfly, Athalia rosae” reports an RNAi experiment repeatedly injecting dsx dsRNA into developing diploid male sawflies, resulting in total feminization of reproductive morphology and behavior. The experiment builds nicely on a previous result from the authors, wherein dsx knockdown in embryos only resulted in feminization of morphology but not female reproductive success.  I’ll also note that the paper is very well written, and I especially enjoyed reading the discussion. Taken at face value, this would be a great study with potential to substantially advance our understanding of the evolution of sex-determining mechanisms in insects. Unfortunately, there are so many details missing in the methods, unreported results, and missing controls that it is very difficult to have confidence in these results. There are also several aspects of the manuscript preparation that are simply unprofessional. Therefore, I cannot recommend acceptance without seeing a massive overhaul of the manuscript and its data.

Major comments:

1. I am really surprised to not see any data from a control group of either non-injected diploid males or mock injected (with dsRED or saline, etc.) diploid males. The lack of this control causes several difficulties for the interpretation of the results. First and foremost, we cannot actually assess whether the dsRNA treatment actually knocked down dsx This is a very standard component of RNAi studies that verifies that manipulation of the target gene is actually what is causing the change in phenotype, not off-target effects or other changes due to injection.

We also simply need to see non-injected diploid males so that we can be confident that they themselves do not display any feminized phenotypes. That would of course undercut the authors conclusions, but right now the reader doesn’t see any data supporting the assumption that diploid males express complete maleness. This criticism might seem foolish to the authors, but given haplodiploid inheritance it’s certainly not implausible that diploidy in males could induce a degree of feminization – actually, the authors make this point themselves (lines 391-395; 424-426).

The bottom line is that these are key controls, without which we cannot have confidence in the conclusions of the study.

2. We need to see baseline data on the sample sizes and results of the dsRNA experiments. How many pupae were initially injected, how many survived, how many offspring did they produce, how many of those were haploid males, how many were diploid males, how many were females.  Did every injected individual display the feminized phenotype or were there diploid males that looked and acted like males? How many individuals were used in the datasets reported in Figures 1-3? It is impossible to assess the study unless the authors are more transparent.

3. There needs to be much more clarity on how sex determination of diploids was performed. Lines 123-125 and Figure S2 suggest that sex determination of diploids was made well after the fact by using PCR and electrophoresis to verify production of the male and female forms of dsx. Elsewhere (lines 236-239), the authors suggest that diploid males were screened and selected as pupae, which as far as I can tell would be impossible based on the description in Figure S1 (diploidy can be determined but not sex). If it is true that sex was not determined until 7+ days after adult emergence, this raises a number of questions. How could the authors have done the experiment in Figure 1C? Was every individual PCR tested to see which dsx form was produced (Figure S2B only shows 12 individuals)?

4. Several experimental details and even whole experiments are not described in the methods, but are rather left out (see also comment 4 below) or described in the results. The experiments on haploid males are the biggest example of this. All work needs to be described in the methods section.

5. It is not appropriate to present the results of an entire experiment as an appendix that is only referenced in the discussion. The authors either need to include the experiment on fruitless in their methods and results like they would any other experiment, or they need to leave it out of the manuscript entirely.

Minor comments:

Line 13: Sawflies are no more a “primitive species of holometabolous insects” any more than any other hymenopteran, please revise. See also comment on Figure 5 below.

Line 114: What level of humidity is “sufficient humidity”?

Lines 144-148: Although the authors cite several previous papers, I think a bit more in the text about injection procedures would be appropriate here given the importance of those procedures to understanding the results.

Lines 162, many others:  Please explain what “naïve” males are.

Lines 192-193: If the results of the ARdsx2 dsRNA treatment are relevant to the conclusions of the manuscript then they need to be shown, at least in supplementary material. We should not be expected to just take the authors word for it that they replicated their results.  If the authors don’t want to include their results, they do not need to be mentioned here or in Figure 1B.

Figure 1: Why does this figure statistically compare control gene expression to dsx expression? The purpose of the control gene is to standardize measurement of the target gene across samples – which the authors don’t even do. The direct comparison of control gene to target gene is meaningless. A better designed experiment could have quantitatively compared dsx expression between control diploid males, treated diploid males, and females across life stages.

Figure 5: The authors might note when discussing this figure and the interpretation stemming from it (lines 299-320) that if indeed dsx is only required for male development in sawflies then a role for dsx in female development evolved independently at least twice – once within the hymenoptera and once in the clade containing the rest of the holometabolous insects. It would also be possible (maybe more likely) that the female development role of dsx was secondarily lost in sawflies. Neither would be impossible, of course, but neither are parsimonious and thus likely not what many researchers would have expected, and would therefore be noteworthy.

Author Response

Our alterations as a result of the reviewer’s comments are as follows.

Reviewer #1

Major comments:

  1. I am really surprised to not see any data from a control group of either non-injected diploid males or mock injected (with dsRED or saline, etc.) diploid males. The lack of this control causes several difficulties for the interpretation of the results. First and foremost, we cannot actually assess whether the dsRNA treatment actually knocked down dsx This is a very standard component of RNAi studies that verifies that manipulation of the target gene is actually what is causing the change in phenotype, not off-target effects or other changes due to injection.

→We thank for the reviewer's comments. Since the phenotype of the negative control individual of the diploid male was exactly the same as that of the normal male (haploid male), we inadvertently forgot to present the diploid male and the haploid male separately. rice field. We apologize for this. We fully agree with the reviewer's opinion. Following the reviewer's comments, we have added data from non-injected diploid males and mock injected (with EGFP dsRNA) diploid males to the revised manuscript. See also Fig. 2E and 2H and 3E and 3H, and Fig, 4 in the revised manuscript.

  1. We also simply need to see non-injected diploid males so that we can be confident that they themselves do not display any feminized phenotypes. That would of course undercut the authors conclusions, but right now the reader doesn’t see any data supporting the assumption that diploid males express complete maleness. This criticism might seem foolish to the authors, but given haplodiploid inheritance it’s certainly not implausible that diploidy in males could induce a degree of feminization – actually, the authors make this point themselves (lines 391-395; 424-426).

The bottom line is that these are key controls, without which we cannot have confidence in the conclusions of the study.

 →As mentioned above, we have added multiple data from non-injected diploid males to the revised manuscript (please also see Fig. 2E and 3E). You can see from those figures that the sexual traits of diploid males are the same as those of normal haploid males. In addition, the quantitative evaluation data of sexual behaviors shown in Fig. 5 supports that the sexual behavior exhibited by diploid males shows the same level of activity as normal males (haploids). Furthermore, past literature has also reported that diploid males show fertility as normal males and produce triploid offspring by mating with females (diploids) (Naito and Suzuki, 1991). This article is cited in our manuscript. Therefore, it is certainly unlikely that diploid males will show femaleness. From the above, it can be concluded that the main reason why Ardsx KD diploid males showed femaleness was not because they were diploid, but because their expression of Ardsx was disrupted. However, as we mentioned in the manuscript, I think that diploidy is, albeit apparently, important for a more thorough sex-reversal. In other words, it is most reasonable to assume that further femaleness caused by diploidy is elicited by disrupting Ardsx expression. We've added these points to the revised manuscript discussion section, to avoid confusing the reader. Please also see lines 498-501.

  1. We need to see baseline data on the sample sizes and results of the dsRNA experiments. How many pupae were initially injected, how many survived, how many offspring did they produce, how many of those were haploid males, how many were diploid males, how many were females.  Did every injected individual display the feminized phenotype or were there diploid males that looked and acted like males? How many individuals were used in the datasets reported in Figures 1-3? It is impossible to assess the study unless the authors are more transparent.

→We would like to thank the reviewers for their comments. Unfortunately, we are unable to present all the data requested by the reviewers this time, because we did not collect those data. We apologize for this. However, we were able to create a table containing the number of individuals tested for injection, the number of individuals examined at the adult stage, and the number of them that showed sex reversal in their sexual phenotype. We've added it to the revisec manuscript. Please also see the data in Table 1.

  1. There needs to be much more clarity on how sex determination of diploids was performed. Lines 123-125 and Figure S2 suggest that sex determination of diploids was made well after the fact by using PCR and electrophoresis to verify production of the male and female forms of dsx. Elsewhere (lines 236-239), the authors suggest that diploid males were screened and selected as pupae, which as far as I can tell would be impossible based on the description in Figure S1 (diploidy can be determined but not sex). If it is true that sex was not determined until 7+ days after adult emergence, this raises a number of questions. How could the authors have done the experiment in Figure 1C? Was every individual PCR tested to see which dsx form was produced (Figure S2B only shows 12 individuals)?

 →As reviewers fear, we were not able to quantify the expression level of Ardsx in diploid males in this study. This is because, as the reviewer pointed out, it is not possible to recognize whether a diploid male is a genetic male without looking at the results of RT-PCR performed at least 7 days after emergence. Morphological and sexual behavioral analyses were all performed without knowing the sex of the diploid. After all experiments were completed, we finally determined which data were from diploid females and which results were from diploid males, based on the results of molecular sexing by RT-PCR. This task was extremely cumbersome, but at this time there was no other way because there is no visible sex-linked marker gene in the sawfly. We hope that the reviewer will understand this point. From a different point of view, the data obtained in this study can be considered reliable because all experiments were conducted under the blind condition of not knowing which individual was female and which was male. Since RNAi suppresses the expression of Ardsx in diploid females, we predict that similar RNAi will significantly reduce the expression of Ardsx in diploid males as well. We have added explanations for these points in the discussion section. Pleas also see lines 481-487.

  1. Several experimental details and even whole experiments are not described in the methods, but are rather left out (see also comment 4 below) or described in the results. The experiments on haploid males are the biggest example of this. All work needs to be described in the methods section.

 →We apologize for the duplicate explanation. Following the reviewer's comments, the lines 241-245 in the results section (explanations about how to make diploid males) has been removed since the same explanations are present in the Materials and Methods section.

  1. It is not appropriate to present the results of an entire experiment as an appendix that is only referenced in the discussion. The authors either need to include the experiment on fruitless in their methods and results like they would any other experiment, or they need to leave it out of the manuscript entirely.

 →According to the reviewer's comment, we decided to remove the fru experiment and its results because these results are not directly related to our research.

Minor comments:

  1. Line 13: Sawflies are no more a “primitive species of holometabolous insects” any more than any other hymenopteran, please revise. See also comment on Figure 5 below.

→According to the reviewer's comments, the words "primitive species" have been all removed from our manuscript.

  1. Line 114: What level of humidity is “sufficient humidity”?

→The level of humidity described by "sufficient humidity" is more than 90 % relative humidity. We have added this in line 121.

  1. Lines 144-148: Although the authors cite several previous papers, I think a bit more in the text about injection procedures would be appropriate here given the importance of those procedures to understanding the results.

→The following explanations have been added to the Materials and Methods section in the revised manuscript. "Female pupae at three days after pupation were subjected to parental RNAi. dsRNAs were injected at the suture between third and fourth abdominal segments of a pupa. To achieve knockdown of Ardsx expression throughout the entire lifespan, further injections of dsRNA into progeny obtained via parental RNAi were performed at the second, third, and final instar larval stages. When the larvae were subjected to RNAi, dsRNAs were injected into the dorsal hemocoel in the second abdominal segment of a larvae. Approximately 1 uL of dsRNA solution with concentration of 100 ng dsRNA /uL was injected into each animal." Please also see lines 166-176.

  1. Lines 162, many others:  Please explain what “naïve” males are.

→Sorry for the lack of explanation. A naïve male is a male who has never met a female and has never experienced mating. We have added this description to the revised manuscript. Please also see lines 191-192 and lines 35-353.

  1. Lines 192-193: If the results of the ARdsx2 dsRNA treatment are relevant to the conclusions of the manuscript then they need to be shown, at least in supplementary material. We should not be expected to just take the authors word for it that they replicated their results.  If the authors don’t want to include their results, they do not need to be mentioned here or in Figure 1B.

→According to the reviewer's comments, we have decided not to mention the results of Ardsx2 dsRNA treatment.

  1. Figure 1: Why does this figure statistically compare control gene expression to dsx expression? The purpose of the control gene is to standardize measurement of the target gene across samples – which the authors don’t even do. The direct comparison of control gene to target gene is meaningless. A better designed experiment could have quantitatively compared dsx expression between control diploid males, treated diploid males, and females across life stages.

→The results shown in Figure 1 are the amount of Ardsx mRNA in individuals injected with Ardsx dsRNA and negative control individuals injected with EGFP dsRNA.We have not quantified the expression level of the EGFP gene.I am sorry that the graph is difficult to understand.The graph has been modified to avoid such misunderstandings.Please also see Fig. 1C.For the reasons mentioned above, we were not able to quantify the expression level of Ardsx in diploid males.

  1. Figure 5: The authors might note when discussing this figure and the interpretation stemming from it (lines 299-320) that if indeed dsx is only required for male development in sawflies then a role for dsx in female development evolved independently at least twice – once within the hymenoptera and once in the clade containing the rest of the holometabolous insects. It would also be possible (maybe more likely) that the female development role of dsx was secondarily lost in sawflies. Neither would be impossible, of course, but neither are parsimonious and thus likely not what many researchers would have expected, and would therefore be noteworthy.

 →We would like to express our sincere gratitude for the reviewer's important comments. We agree with the reviewer's suggestion that the female developmental role of dsx may have been secondarily lost in sawfly. I think the false prejudice that sawfly is the most primitive species has biased the insights derived from our results. Following the reviewers' comments, we have partially modified or added the text in the discussion section. Please also see lines 398-402.

Reviewer 2 Report

The manuscript by Suzuki and co-workers describes the effect of doublesex mRNA knockdown on the development of male sawfly, Athalia rosae. They used the dsRNA injection approach to selectively deplete dsx mRNA for the entire life span by repeated dsRNA injection and then examined the sexual differentiation, oogenesis, and sexual behavior. They claim that dsx is essential for male development in sawfly and its depletion leads to complete male to female sex reversal.

However, a similar phenotype has already been reported by selective KD of Ardsx during the early pupal stage (Mine et al, 2017). This manuscript is just an extension of a part of that study and has not made any significant progress in our understanding of sex determination in sawfly. The authors have not provided any details of the mode of action of dsx and other factors in sex determination and oogenesis. In short, the manuscript is not up to the mark to be published in the journal Insects and needs additional data to support the present study and define the mechanism of sex determination and gametogenesis in sawfly.

Specific Points:

  1. What is the mechanism of action of dsx in sex determination in sawfly? What are the other factors involved in this process?
  2. All the observations are based on the assumption that partial depletion of dsx mRNA will finally cause a significant depletion of dsx protein, but this has never been validated experimentally.
  3. Immunostaining or western blot against dsx protein will provide a better understanding of its mode of action.
  4. The RT-PCR data in the graph has not been normalized which makes the analysis not very clear.

Author Response

Our alterations as a result of the reviewer’s comments are as follows.

Reviewer #2

The manuscript by Suzuki and co-workers describes the effect of doublesex mRNA knockdown on the development of male sawfly, Athalia rosae. They used the dsRNA injection approach to selectively deplete dsx mRNA for the entire life span by repeated dsRNA injection and then examined the sexual differentiation, oogenesis, and sexual behavior. They claim that dsx is essential for male development in sawfly and its depletion leads to complete male to female sex reversal.

However, a similar phenotype has already been reported by selective KD of Ardsx during the early pupal stage (Mine et al, 2017). This manuscript is just an extension of a part of that study and has not made any significant progress in our understanding of sex determination in sawfly. The authors have not provided any details of the mode of action of dsx and other factors in sex determination and oogenesis. In short, the manuscript is not up to the mark to be published in the journal Insects and needs additional data to support the present study and define the mechanism of sex determination and gametogenesis in sawfly.

Specific Points:

  1. What is the mechanism of action of dsx in sex determination in sawfly? What are the other factors involved in this process?

→Sorry for the lack of explanation about the function of dsx ortholog in the sawfly (designated as Ardsx). Similar to dsx genes so far identified in other holometabolous insects, Ardsx pre-mRNA was spliced alternatively in a sex-dependent manner, yielding female- and male-specific isoforms (described as ArdsxF and ArdsxM, respectively). Our previous findings demonstrate that ArdsxM is required for male development (Mine et al., 2017). Thus, like dsx orthologs in other holometabolous insects, Ardsx is likely to act at the bottom of the sex determination cascade and regulates sexual development of the sawfly. We have added these explanations in introduction section of the revised manuscript. Please also see lines 85-90.

  1. All the observations are based on the assumption that partial depletion of dsx mRNA will finally cause a significant depletion of dsx protein, but this has never been validated experimentally.

→Actually, we also wanted to examine the expression of Ardsx at the protein level, but since there is no antibody against this protein, it is not possible to examine the expression of ArDSX protein by methods such as western blotting and immunostaining at this time. I've added the following text to the revised manuscript to draw the reader's attention to this point:

 “A caveat is that we were not able to confirm that the expression of Ardsx was reduced at the protein level because there were no antibodies that specifically recognize the ArDSX protein. Therefore, the absence of abnormalities in female development in Ardsx RNAi treated females may simply reflect insufficient level of knockdown of the ArDSX protein expression in females. However, it is unlikely that ArDSX protein level was not reduced only females since the extent of reduction of Ardsx mRNA by our RNAi was almost similar in both males and females.” Please also see lines 370-376.

  1. Immunostaining or western blot against dsx protein will provide a better understanding of its mode of action.

→We fully agree with the reviewer's comments.However, as mentioned above, since there is no antibody that specifically recognizes the ArDSX protein at this time, it is not possible to investigate the expression pattern of the ArDSX protein and its localization by immunostaining or western blot.We would like to make these studies a future subject.

  1. The RT-PCR data in the graph has not been normalized which makes the analysis not very clear.

→The expression level of Ardsx mRNA was standardized by the expression level of the elongation factor-1α gene. I've already mentioned that in the Materials and methods section, but I'm sorry it's hard to understand. The description has been changed to make it easier to understand. Please also see lines 149-152.

Reviewer 3 Report

This manuscript describes the effects of RNAi knockdown of doublsesex in A. rose at multiple stages during the lifecycle. This study effectively extends a previous dsx RNAi report by this group from 2017 (reference 23 in this manuscript). The research reported here is mostly well done. Sex determination in insects is an important area of developmental genetics, and this report would find an interested audience. In general, I am in favor of its publication. However, there are several rather critical issues with the description of methods, the use of statistics, and the misuse of terminology that must be resolved before publication.

The manuscript repeatedly refers to sawflies as a "primitive species". As a term, "primitive" should only be applied to traits, not to species or organisms at any taxonomic level. A species with several primitive traits may nevertheless have many other traits that are derived. Similarly, describing a taxon as "basal" is also problematic, although I acknowledge it is commonly done. For a useful summary of the reasons for these objections, I would point the authors to this blog post http://for-the-love-of-trees.blogspot.com/2016/09/the-ancestors-are-not-among-us.html and encourage them to revise their discussions of evolutionary relationships with "tree thinking" more in mind.

The major contribution of this study is the application of dsx RNAi at two points during the sawfly life cycle: during oogenesis via injection into females and in again into the resulting larvae. I appreciate that this is a technically difficult to do. However, it is overreaching to suggest that this knocks out dsx activity during "the entire lifespan" of the insect or to call it "complete RNAi". I would not object if dsx transcript levels were shown via RT-PCR to be effectively zero at multiple stages (embryo, larval, pupa and adult), but in the absence of such supporting evidence, more conservative terminology must be used to describe these experiments.

There are several points in the Methods section were more detail is necessary. Even where this study extends previous work, I encourage the authors to give critical details about methods. For real-time PCR, it is absolutely essential that the manuscript report the number of biological and technical replicates used. Does the real-time PCR use the delta-delta-Ct method? Or does it include an estimate actual primer efficiencies? (which would be ideal) Were melt curves examined to confirm that primers do not produce off-target products? For RNAi, note the location of injection on the animal, the volume and concentration of the dsRNA and the carrier solution. For the behavioral methods, please cite a previously published description of normal sawfly courtship. The Methods section should include a subsection on statistics! Were tests done by hand? Fine if that's the case, but please cite any software used. -- How were the splice variants of fruitless, shown in figure A1, identified? By RACE?

The quantitative data used to assess phenotypes in this study is a strength. However, there are some issues with reporting and statistics for these data. First, it must be made clear what the sample size is for each group (Figures 1C, 3L, 4A-F). Calculations of mean and standard deviation for a small sample size cannot be reliable. While I know bar plots are still common in the literature, I suggest instead plotting the individual values as dots and the median as a heavy horizontal bar. In this study, the authors make comparisons using t-tests (either what I assumer to be Students' t-test and Welsh's t-test). The assumptions of t-tests (large sample size, generally considered >25; normal distribution; equal variance) make it a poor choice for the data in this study. In most cases, a more appropriate test would be Wilcoxon's rank sum test (WRST). A convenient implementation of this test exist in base R.

Across the figures, it would be helpful to have a consistent order for the sexes. e.g. female first. Figures 1-3 alternate which sex is shown first! I also suggest including the Greek male and female symbols.

In Figure 2, it seems necessary to include control diploid male anatomy. I would also crop in more on the genitalia and perhaps make the panels taller, so that they appear larger on the page.

While it may be beyond the scope of this study, it might be interesting for the authors to consider examining the anatomy of female and "pseudofemale" genitalia using more sensitive methods, such as landmark-based geometric morphometrics or Fourier analysis.

Instead of the "data not shown" statement in line 333, stating that embryonic dsx RNAi resulted in males lacking gonads, this phenotype should be shown in a figure! How many individuals did this occur in? And while the text says "embryonic knockdown", is this actually referring to parental RNAi?

The presentation of the fruitless data in the Discussion should be moved to the Results section. However these results are strange. I don't see any obvious differences in expression in males vs. females. How should that fact be interpreted? How could this gene promote male-specific behavior if it's also expressed in females?

The Discussion ends with speculation regarding the cause of female fertility in "pseudofemale" dsx RNAi individuals who are genetically male. This is useful. But I wonder if the authors are missing a simple potential explanation based on gene dosage. The differentiation of many female phenotypes, well downstream of dsx, may simply require two copies of effector genes for proper development.

Finally, it is puzzling that a female-specific transcript of dsx exists in this species if it is not functional required for female development. This observation is inconsistent with the hypothesis that sawflies represent a primitive "male-only" requirement for dsx. Perhaps the Discussion can consider this.

Minor concerns

With reference to the manuscript's line numbers below.

  • 58: Preface this section by stating, "In Drosophila melanogaster..."

  • 64: "double sexuality" is a confusing way to describe this phenotype.

  • 87: Please note here that RNAi was done during the larval (or pupal?) stage.

  • 114: Please note the percent relative humidity.

  • 120: should be "were subjected to"

  • 125: Unclear. Please re-phrase.

  • 197, "expected duration": I don't see how the duration of RNAi can be known. Cut our provide direct evidence in support.

  • 223: Throughout the manuscript I believe "internal genitalia" are actually the gonads.

  • 228: Starting on this line, it seems appropriate to start a new subsection.

  • 243: It's very confusing to refer to these individuals as "males". At the very end of the Results, the term "pseudofemale" is introduced. I'm not sure if this is quite right, but I suggest using a consistent term for csd-homozygous diploid individuals (genetic males?) who display female phenotypes in sexually dimorphic traits.

  • 249: RNAi at what stage or stages?

  • 268: The wording here regarding "extent of femaleness" is very confusing. Please revise.

  • 302-303: Cladocera (by themselves) are not the sister group to insects. There are several other non-insect arthropod groups with closer relationships. Please revise.

  • 370-371, "sexual orientation": like "gender", "sexual orientation" is a term best reserved for human behavior. Please revise.

  • 383, "nearly all": Confusing. Please revise.

  • 391: Preface this statement with, "In A. rose..."

  • 408: Preface this statement with, "In D. melanogaster..."

  • 408, "double-switch gene": This is an odd term. At least here. Please revise.

  • Reference 5 was published in 2013

Author Response

Our alterations as a result of the reviewer’s comments are as follows.

Reviewer #3

This manuscript describes the effects of RNAi knockdown of doublsesex in A. rose at multiple stages during the lifecycle. This study effectively extends a previous dsx RNAi report by this group from 2017 (reference 23 in this manuscript). The research reported here is mostly well done. Sex determination in insects is an important area of developmental genetics, and this report would find an interested audience. In general, I am in favor of its publication. However, there are several rather critical issues with the description of methods, the use of statistics, and the misuse of terminology that must be resolved before publication.

The manuscript repeatedly refers to sawflies as a "primitive species". As a term, "primitive" should only be applied to traits, not to species or organisms at any taxonomic level. A species with several primitive traits may nevertheless have many other traits that are derived. Similarly, describing a taxon as "basal" is also problematic, although I acknowledge it is commonly done. For a useful summary of the reasons for these objections, I would point the authors to this blog post http://for-the-love-of-trees.blogspot.com/2016/09/the-ancestors-are-not-among-us.html and encourage them to revise their discussions of evolutionary relationships with "tree thinking" more in mind.

  1. The major contribution of this study is the application of dsx RNAi at two points during the sawfly life cycle: during oogenesis via injection into females and in again into the resulting larvae. I appreciate that this is a technically difficult to do. However, it is overreaching to suggest that this knocks out dsx activity during "the entire lifespan" of the insect or to call it "complete RNAi". I would not object if dsx transcript levels were shown via RT-PCR to be effectively zero at multiple stages (embryo, larval, pupa and adult), but in the absence of such supporting evidence, more conservative terminology must be used to describe these experiments.

→We would like to express our sincere gratitude to the reviewers for their valuable comments. We have withdrawn "the entire life span" and all similar expressions. Instead, we have changed it to "during several developmental stages" or "to prolong the duration of the RNAi-mediated mRNA knockdown" or "repeated injections of dsRNAs". At the same time, we have deleted all the expressions "complete RNAi". Instead, we used the expression "repeated injections of dsRNAs".

  1. There are several points in the Methods section were more detail is necessary. Even where this study extends previous work, I encourage the authors to give critical details about methods. For real-time PCR, it is absolutely essential that the manuscript report the number of biological and technical replicates used. Does the real-time PCR use the delta-delta-Ct method? Or does it include an estimate actual primer efficiencies? (which would be ideal) Were melt curves examined to confirm that primers do not produce off-target products? For RNAi, note the location of injection on the animal, the volume and concentration of the dsRNA and the carrier solution. For the behavioral methods, please cite a previously published description of normal sawfly courtship. The Methods section should include a subsection on statistics! Were tests done by hand? Fine if that's the case, but please cite any software used. -- How were the splice variants of fruitless, shown in figure A1, identified? By RACE?

→We would like to thank the reviewers for their very detailed comments. We apologize for the lack of explanation about the materials and methods.

    The real-time PCR used the delta-delta-Ct method. We checked dissociation curves of qPCR reactions to confirm that primers do not produce off-target products. We have added these explanations into the Materials and Methods section in the revised manuscript. Please also see lines 147-155.

    For RNAi, female pupae at three days after pupation were subjected to parental RNAi. In this case, dsRNAs were injected at the suture between third and fourth abdominal segments of a pupa. To achieve knockdown of Ardsx expression in order to prolong the duration of the RNAi-mediated mRNA knockdown, further injections of dsRNA into progeny obtained via parental RNAi were performed at the second, third, and final instar larval stages. When the larvae were subjected to RNAi, dsRNAs were injected into the dorsal hemocoel in the second abdominal segment of a larvae. Approximately 1 uL of dsRNA solution with concentration of 100 ng dsRNA /uL was injected into each animal. We have added these explanations into the Materials and Methods section in the revised manuscript. Also please see lines 165-176.

  For the behavioral methods, we have cited the following previously published papers describing normal sawfly courtship.

Amano, T., Nishida, R., Kuwahara, Y., and Fukami, H. (1999). Pharmacophagous acquisition of clerodendrins by the turnip sawfly (Athalia rosae ruficornis) and their role in the mating behaviour. Chemoecology 9, 145–150.

Awane K, Kitano H (1992) Studies on the effect of Clerodendron trichotomum Thunb. on the mating behavior of Athalia infu- mata (Marlatt) (Hymenoptera: Tenthredinidae). Jpn J Appl Entomol Zool 36:13–16

Kitano H (1988) Experimental studies on the mating behavior of Athalia lugens infumata. Kontyu 56:180–188

Please also see references 27-29.

Some of the above papers mainly investigated the effect of pharmacophagous acquisition of clerodendrins on sawfly mating, but also included a description of sawfly's normal courtship behavior. The typical courtship behavior of the sawfly based on previous reports is as follows: 1) the male sawfly explores the female in a waiting position. 2) when the male visually recognizes the female, the male chases the female, and touches and licks the female's body surface, 3) then the male uses the tail clasper to grab the female's copulatory organ and insert his penis (defined as copulation), 4) the male and the female stay still for a few to approximately 30 min in a tail-to-tail connection (defined as copulation success), 5) when the male complete spermatophore transfer to the female, then the male leaves the female. We have added these explanations into the Materials and Methods section in the revised manuscript. Please also see lines 199-206.

    We have included a subsection on statistics in the Materials and Methods section of the revised manuscript. All statistical processing was performed using Easy R (EZR) software (https://www.jichi.ac.jp/saitama-sct/SaitamaHP.files/download.html). The Shapiro-Wilk test was used to evaluate whether the data obtained by our experiments showed a normal distribution. Since the number of samples was less than 25 and did not show a normal distribution, the significant difference test between the two groups was performed using Mann-Whitney U test. Each data obtained in this study was displayed in box plot format. The box plot was created using the EZR software with default setting. Pleas also see lines 212-219.

 According to the other reviewer's comments (comments from reviewer #1), we have removed the results of RT-PCR analysis for fru expression.

  1. The quantitative data used to assess phenotypes in this study is a strength. However, there are some issues with reporting and statistics for these data. First, it must be made clear what the sample size is for each group (Figures 1C, 3L, 4A-F). Calculations of mean and standard deviation for a small sample size cannot be reliable. While I know bar plots are still common in the literature, I suggest instead plotting the individual values as dots and the median as a heavy horizontal bar. In this study, the authors make comparisons using t-tests (either what I assumer to be Students' t-test and Welsh's t-test). The assumptions of t-tests (large sample size, generally considered >25; normal distribution; equal variance) make it a poor choice for the data in this study. In most cases, a more appropriate test would be Wilcoxon's rank sum test (WRST). A convenient implementation of this test exist in base R.

→We thank for the valuable reviewer's comments. The sample size for each group is described below each box plot. As mentioned above, the significant difference between the two groups was re-examined using the Mann-Whitney U test provided by Easy R software. Please also see the subsection on statistics in the Materials and Methods section of the revised manuscript.

  1. Across the figures, it would be helpful to have a consistent order for the sexes. e.g. female first. Figures 1-3 alternate which sex is shown first! I also suggest including the Greek male and female symbols.

→ We are very sorry that the order of displaying female data and male data is not unified.For all data, we have unified so that the female data is shown first.

  1. In Figure 2, it seems necessary to include control diploid male anatomy. I would also crop in more on the genitalia and perhaps make the panels taller, so that they appear larger on the page.

→According to the reviewer's comment, we have included control diploid anatomy. Since Figure 2 was enlarged as much as possible in the revised manuscript, we think that the reader can see the detail of the genital structures without cropping them.

  1. While it may be beyond the scope of this study, it might be interesting for the authors to consider examining the anatomy of female and "pseudofemale" genitalia using more sensitive methods, such as landmark-based geometric morphometrics or Fourier analysis.

→We thank for the reviewer's helpful comments. We would like to carry out the analysis using methods that the reviewer points out in the future. However, at this time, we have not yet mastered methods such as landmark-based geometric morphometrics or Fourier analysis, so we would like to just present the photographs.

  1. Instead of the "data not shown" statement in line 333, stating that embryonic dsx RNAi resulted in males lacking gonads, this phenotype should be shown in a figure! How many individuals did this occur in? And while the text says "embryonic knockdown", is this actually referring to parental RNAi?

→We have deleted the results of parental RNAi experiment.As mentioned in the introduction section, the main purpose of this study is to suppress Ardsx expression with repeated injections of dsRNA for as long as possible.In contrast, the result of parental RNAi is only the result of a transient knockdown.The parental RNAi experiment in this paper is kind of a preliminary experiment that confirmed that parental RNAi works in this study as well.For these reasons, we have removed the results for parental RNAi.

  1. The presentation of the fruitless data in the Discussion should be moved to the Results section. However these results are strange. I don't see any obvious differences in expression in males vs. females. How should that fact be interpreted? How could this gene promote male-specific behavior if it's also expressed in females?

→According to the other reviewer's comments (comments from reviewer #1), we have removed the results of RT-PCR analysis for fru expression.

  1. The Discussion ends with speculation regarding the cause of female fertility in "pseudofemale" dsx RNAi individuals who are genetically male. This is useful. But I wonder if the authors are missing a simple potential explanation based on gene dosage. The differentiation of many female phenotypes, well downstream of dsx, may simply require two copies of effector genes for proper development.

→We agree with the idea that the differentiation of many female phenotypes may simply require two copies of effector genes for proper development. We have added this idea to the end of the discussion section in the revised manuscript. Pleas also see lines 497-498.

  1. Finally, it is puzzling that a female-specific transcript of dsx exists in this species if it is not functional required for female development. This observation is inconsistent with the hypothesis that sawflies represent a primitive "male-only" requirement for dsx. Perhaps the Discussion can consider this.

→We are also wondering why the bee produces a female dsx transcript if the female dsx has no function. In fact, we initially suspected that a female knocked down by Ardsx did not show any abnormalities. However, this result showed extremely high reproducibility. Moreover, several reports supporting our results have been published in recent years. As mentioned in the discussion section, one of the hemimetabolous insects, German cockroach, Blattella germanica, produces female-specific dsx transcripts, but it is not required for female differentiation. Ledo ́n-Rettig et al. hypothesized that sex-specific splicing of dsx in insects is derived from the non-spliced ancestral condition, and thus, the female DSX isoform is relatively novel and does not contribute largely to female development in several tissues. These findings suggest that our conclusion that the female isoform of dsx transcript is non-functional in the sawfly is not so mysterious. Further identification and functional analysis of dsx in other Hymenopteran species will validate our conclusions. Please also see lines 377-401.

Minor concerns

With reference to the manuscript's line numbers below.

  1. Line 58: Preface this section by stating, "In Drosophila melanogaster..."

→We have added "In Drosophila melanogaster" to the position where the reviewer pointed. Please also see line 56.

  1. Line 64: "double sexuality" is a confusing way to describe this phenotype.

→We have removed "double sexuality" and replaced it with "intersexuality" in the revised manuscript. Please also see line 63 in the revised manuscript.

  1. Line 87: Please note here that RNAi was done during the larval (or pupal?) stage.

→We have inserted "during pupal stage" to the position where the reviewer pointed. Please also see line 91 in the revised manuscript.

  1. Line 114: Please note the percent relative humidity.

→We have noted the percent relative humidity. Please also see line 121 in the revised manuscript.

  1. Line 120: should be "were subjected to"

→We have changed "was subjected to" to "were subjected to" according to the reviewer's comment. Please also see line 121 in the revised manuscript.

  1. Line 125: Unclear. Please re-phrase.

→We have added more detailed explanations into the position where the reviewer pointed. Please also see lines 133-136 in the revised manuscript.

  1. Line 197, "expected duration": I don't see how the duration of RNAi can be known. Cut our provide direct evidence in support.

→We have deleted all of the "expected duration" as a result of the modification of Figure 1 according to the other reviewer's comment. Please also see the legend of Figure 1 in the revised manuscript.

  1. Line 223: Throughout the manuscript I believe "internal genitalia" are actually the gonads.

→The organs we examined for observation was the organs commonly defined as the internal genitalia, which include not only testis and ovary but also include other reproductive organs such as spermatheca, uterus, accessory gland, ejaculatory duct, seminal vesicle, and vas deferens. However, as the reviewer pointed, male-to-female sex reversal was observed especially in the gonads. Therefore, we have replaced some parts described "internal genitalia" with gonads. Please also see lines 94, 106, 254, 301, and 312.

  1. Line 228: Starting on this line, it seems appropriate to start a new subsection.

→According to the reviewer's comment, we have created a new subsection, which focuses on the properties and fertility of eggs found in the ovaries of Arsx KD males. Pleas also see lines 257-313.

  1. Line 243: It's very confusing to refer to these individuals as "males". At the very end of the Results, the term "pseudofemale" is introduced. I'm not sure if this is quite right, but I suggest using a consistent term for csd-homozygous diploid individuals (genetic males?) who display female phenotypes in sexually dimorphic traits.

→We apologize for the confusion. At the moment, the expression “pseudofemale” is the most reasonable and I think it is the most appropriate expression for the reader to understand the context. For convenience only, we would like to refer to Ardsx knockdown diploid males, which indicate male-to-female sex reversal, as pseudofemale in the revised manuscript. We have added "pseudofemale" to the place where appropriate for the reader to understand our results. Pleas also see lines 310-313, 329-342, 344, 347, 352, 353, and revised version of Figure 5 and its legend.

  1. Line 249: RNAi at what stage or stages?

→Sorry for the lack of explanation. Since the experimental results of parental RNAi have been deleted in the revised manuscript, all RNAi in the revised manuscript correspond to RNAi that combines parental RNAi and repeated injections of dsRNA at 2nd, 3rd, last instar larval stages.

  1. Line 268: The wording here regarding "extent of femaleness" is very confusing. Please revise.

→Sorry for the explanation being confusing. We have removed the sentence including this confusing phrase.

  1. Lines 302-303: Cladocera (by themselves) are not the sister group to insects. There are several other non-insect arthropod groups with closer relationships. Please revise.

→We thank the reviewer's comment. The sentence "which are the closest relatives to the insects" has been removed.

  1. Lines 370-371, "sexual orientation": like "gender", "sexual orientation" is a term best reserved for human behavior. Please revise.

→We thank the reviewer's comment. We have changed "sexual orientation" to "sexual identity" in the revised manuscript. Please also see line 427-428.

  1. Line 383, "nearly all": Confusing. Please revise.

→We thank the reviewer's comment. We have changed "nearly all" to "most" in the revised manuscript. Please also see line 444.

  1. Line 391: Preface this statement with, "In A. rose..."

→We have preface this statement with "In A. rosae..." in the revised manuscript. Also please see line 441.

  1. Line 408: Preface this statement with, "In D. melanogaster..."

→We have preface this statement with "In D. melanogaster..." in the revised manuscript. Pleas also see line 470.

  1. Line 408, "double-switch gene": This is an odd term. At least here. Please revise.

→We have removed the phrase "double-switch gene" and re-phrased as follows:

" In D. melanogaster, the dsx gene acts at the bottom of the sex-determination cascade that induces appropriate sexual differentiation in each sex according to upstream genetic sex-determining signals."

Pleas also see lines 470-472.

  1. Reference 5 was published in 2013

→We are sorry for this. We have changed "2018" to "2013" in the revised manuscript.

Round 2

Reviewer 2 Report

I agree with the author's limitation about antibody. To confirm the phenotype directly related with Ardsx downregulation and not because of any offtargets, author should use at least two different RNAi construct.

Author Response

I agree with the author's limitation about antibody. To confirm the phenotype directly related with Ardsx downregulation and not because of any offtargets, author should use at least two different RNAi construct.

Answer: Thank you for understanding our limited situation with antibodies. We agree with the comment "author should use at least two different RNAi construct". In fact, in addition to the Ardsx dsRNA published in the paper submitted this time, we performed RNAi using another dsRNA (Ardsx2 dsRNA) targeting Ardsx mRNA. These two dsRNAs are identical to the dsRNAs used in our previous report. Apparently male-to-female sex reversal was observed in the individuals obtained as a result of RNAi using Ardsx2 dsRNA. However, there are no photographs of the external genitalia because we did not prepare cuticle specimens for those individuals. On the other hand, we were able to find several pictures of the internal genitalia, so we included them in the Supplementary file in Figure S3. Please also see Figure S3.

Reviewer 3 Report

The authors have adequate addressed the concerns raised in my initial review. Several of the figures are rotated 90 degrees. And I noticed only a few spelling and grammar errors. ("Phanotype" in the title to Table 1, and some unnecessary "the"'s.)

Author Response

The authors have adequate addressed the concerns raised in my initial review. Several of the figures are rotated 90 degrees. And I noticed only a few spelling and grammar errors. ("Phanotype" in the title to Table 1, and some unnecessary "the"'s.)

Answer: For the comment "Several of the figures are rotated 90 degrees", we have identified the cause of this issue and made improvements. We have identified some of the spelling and grammar errors that were pointed out and fixed them appropriately. Following the reviewer's comments, we have removed some "the" that we think are unnecessary. See the comments section in the Word file for details.